# A Heart Rate Step Function Response Method for the Evaluation of Pulse Wave Velocity as a Predictor of Major Adverse Cardio-Vascular Events

**DOI:** 10.3390/medicina58111633

**Published:** 2022-11-12

**Authors:** Ioana Marin, Florina Georgeta Popescu, Elena-Ana Pauncu, Adrian Apostol, Viviana Mihaela Ivan, Catalin Nicolae Marin, Ovidiu Fira-Mladinescu, Sorin Ursoniu

**Affiliations:** 1Discipline of Occupational Medicine, Department V—Internal Medicine I, “Victor Babeş” University of Medicine and Pharmacy, Eftimie Murgu Square, No. 2, 300041 Timisoara, Romania; 2Discipline of Cardiology, Department VII—Internal Medicine II, Centre for Molecular Research in Nephrology and Vascular Disease, “Victor Babes” University of Medicine and Pharmacy, Eftimie Murgu Square, No. 2, 300041 Timisoara, Romania; 3Department of Cardiology I, “Pius Brânzeu” County Emergency Clinical Hospital, Liviu Rebreanu Ave., No. 156, 300723 Timisoara, Romania; 4Faculty of Physics, West University of Timisoara, V. Parvan Ave., No. 4, 300223 Timisoara, Romania; 5Discipline of Pulmonology, Department XIII—Infectious Diseases, Centre for Research and Innovation in Precision Medicine of Respiratory Diseases, “Victor Babes” University of Medicine and Pharmacy, Ghe. Adam, No. 13, 300310 Timisoara, Romania; 6Discipline of Public Health, Department of Functional Sciences, Center for Translational Research and Systems Medicine, “Victor Babes” University of Medicine and Pharmacy, Eftimie Murgu Square, No. 2, 300041 Timisoara, Romania

**Keywords:** pulse wave velocity, heart rate, physical effort, step function response

## Abstract

*Background and Objectives*: Cardiovascular diseases are the main cause of death worldwide, and pulse wave velocity (PWV) is considered a predictor of major adverse cardiovascular events. The study intended to be helpful in finding methods for the preliminary assessment of PWV in primary care units. *Materials and Methods*: The study group consisted of 36 subjects (considered healthy by their own statement) from the medical field (medicine students and residents) aged between 20 and 30 years: 33.3% males and 66.7% females. Two types of measurements were carried out successively: (a) measurements with the arteriograph and (b) measurements on a treadmill effort testing system, where heart rate (HR) was measured over time as a response to step function physical effort (PE). *Results*: The study allowed for the highlighting of some limits which, if exceeded, can be associated with high PWV values: (i) if after a moderate PE and a resting time of at least 6 min, the HR is larger than 80 b/min; (ii) if the relaxation time in a PE test of moderate intensity is larger than 1 min; (iii) if the HR measured after the subject is raised from the supine to orthostatic position is larger than 100 b/min, and (iv) if the resting HR is larger than 80 b/min. *Conclusions*: Steady-state HR correlates with PWV and may be used for the preliminary assessment of PWV.

## 1. Introduction

The method of step function response is usually involved in electronic engineering [1] and in materials science [2] for the evaluation of various characteristic quantities or parameters. During any process of changing the state of a system, there is a quantity that characterizes the cause for which the transition is made from one state to another and a quantity that characterizes the effect. The quantity that characterizes the cause is usually called the “stimulus”, and the physical quantity that characterizes the effect is called the “response”. In brief, the method of step function response consists of recording and analyzing the time evolution of a response after the stimulus suddenly changes (from zero to a constant value or from a constant value to zero) [1,2].

For instance, in characterizing the magnetic properties of materials, the stimulus is the intensity of the magnetic field, H, and the response is the magnetization, M, [3]; when characterizing the dielectric properties of materials, the stimulus is the electric field strength, E, and the response is the polarization, P [2].

The concept of step function response can be at least formally extended to any system using a stimulus and a response, for instance, in neuroscience [4,5], the cerebrovascular responses to neural activity [6], or in the analysis of the relationship between the aortic elasticity and urinary arsenic levels [7]. Given that the physiological response of the circulatory system to a physical effort is a change in heart rate, within a formal analysis, we can choose the physical effort as a “stimulus” and the heart rate as a “response” quantity.

Cardiovascular diseases are the main cause of death worldwide, and major adverse cardiovascular events (MACEs) are a common primary outcome of interest [8]. Definition components of MACE include acute myocardial infarction (AMI), coronary syndrome (CS) or ischemic heart disease (IHD), ischemic or hemorrhagic stroke (IS or HS), revascularization procedures, and cardiovascular death [8]. Some of these components may be preceded by arterial stiffness, which is evaluated by measuring the pulse wave velocity (PWV) [9], and that is why the evaluation of PWV is considered a predictor for major adverse cardiovascular events.

There exists dedicated equipment for PWV measurement, but the equipment is only found in some hospitals. For the effective prevention of cardiovascular diseases, even in underdeveloped countries, there should be methods for evaluating PWV, which should be accessible to any general practitioner (GP).

This paper presents a study on the correlation between the measured PWV and some parameters that can be assessed from the heart rate response after the sudden application of a moderate physical effort. The results of the study may be helpful in finding a method for preliminary assessing PWV in primary care units, which will help with the early recognition of susceptibility to cardiovascular diseases.

## 2. Materials and Methods

The study group was selected from the medical field in a county hospital, consisting of 36 subjects, and the inclusion criterion was the statement of each subject that they did not have cardiovascular diseases and were not undergoing treatment for other diseases, with a normal ECG and heart rate response during treadmill effort recording. There were 40 volunteers at the beginning of the study, and four of them were not eligible to continue the study because of different arrhythmias during the physical effort. The study group consisted of nine medicine students (27.3%) that were in the summer practice of the Cardiology Department, with 15 residents in cardiology (42.5%), and 12 residents in diabetes mellitus, nutrition, and metabolic diseases (30.2%), all aged between 20 and 30 years. Of the total number of subjects, 12 (33.3%) were males, and 24 (66.7%) were females. The characteristics of the study group are summarized in Table 1.

All subjects voluntarily agreed to participate in the study, and the informed consent was approved by the Ethics Committee of the University of Medicine and Pharmacy “Victor Babeş”, Timisoara, No. 59/22.12.2021.

The study was based on two types of measurements, which were carried out successively: (a) measurements of pulse wave velocity with the arteriograph and (b) measurements on a treadmill effort testing system.

The stages of the measurements and the time dependence of the physical effort (in arbitrary units) are presented schematically in Figure 1.

(I) Measurements of pulse wave velocity were performed with a TensioMed arteriograph, following the recommendations within [10]: the subjects did not drink alcohol the night before the measurements nor coffee for at least 3 h before the measurements, they did not exercise nor smoke, and they rested for 10 min in a quiet room preceding the measurements. After measuring the arm circumference and the length of the distance between the upper curvature of the sternum and the upper edge of the pubis, PWV measurements were performed in the supine position with the cuff on the dominant arm. At the same time, in this stage, we measured the systolic blood pressure, diastolic blood pressure, and heart rate (denoted by HR-Art). The results measured with the arteriograph are summarized in Table 1;

(II) After performing the measurements with the arteriograph, each subject was raised from the supine position to the orthostatic position and placed on the treadmill. In this position the heart rate was measured at the time moment t_2_ and was denoted by HR-Pre.

Measurements were performed with a Labtech system from Labtech LTD Debrecen, Hungary, on the EC-12 LT Treadmill effort testing system, which was equipped with Cardiospy rest and stress software (EC-12RS PC-based Resting and Stress Test ECG System). The hemodynamic parameters were measured automatically minute by minute. Reports were automatically exported and organized using patient ID and registration time. As an emergency and automatic test stop criterion, we selected the HR value of 160 b/min. During the test, we paid special attention to the heart rate, blood pressure, changes in the ECG pattern, irregular heart rhythm, and the appearance and symptoms of the subjects.

(III) The loading stage was carried out on the treadmill at a speed of 2.7 km/h and lasted for 4 min. For each subject, the heart rate (after 4 min) (at the time moment, t_3_) was measured and denoted by HR-4min;

(IV) The cooling period lasted one minute;

(V) The recovery stage, with the subject in the supine position, lasted 6 min. During the recovery period, from the time moment t_4_ to the time moment t_5_, the heart rate was measured minute by minute and denoted by HR-Relax (t), where t represents time. The heart rate value measured at the end of the recovery period (at the time moment, t_5_) was noted by HR-6min.

The schematic representations were made in Paint, whilst the plots of the results, the fit of the experimental data, and their statistical processing were performed using the OriginPro 8 software (8th version). The means, standard deviations, and Pearson’s correlation coefficients are presented. Linear regression analyses were performed between PWV and the various parameters obtained from the effort stress test, and regression equations were also presented. For correlation, a two-tailed test of significance was used. The statistical significance was set at *p* < 0.05.

## 3. Results

According to the general theory of relaxation processes, if the stimulus, S, changes suddenly (following a step function), then the response, R, of the system has an exponential time dependence [2,3] (schematically represented in Figure 2).

In the case of measurements with the treadmill effort testing system, there are four moments at which the physical effort (as stimulus) changes suddenly (see Figure 1). The step function response of the heart rate was analyzed only for the fourth stage (the recovery stage) due to the fact that it was the longest stage and no other change in effort followed.

The typical time dependence of the heart rate measured on one subject in the recovery stage is shown in Figure 3. The experimental dependence, HR-Relax (t), was fitted with the exponential decay function, y(t) = y_o_ + A∙exp(−t/τ), and, for each subject, the parameters y_o_, A, and τ were determined. The significance of the fit parameters is clear from the expression of y(t) and in Figure 3.

When t tends to infinity, y(t) tends to y_o_, so y_o_ is the resting heart rate value after physical effort. The second parameter, A, is the difference between the heart rate at the beginning of the recovery stage and the resting heart rate after physical effort (A = y(0) − y_o_). The time constant, τ, is the time interval after which the deviation from the resting value of the heart rate (y(t) − y_o_) decreases by *e* times, where *e ≅* 2.71 is the base of the natural logarithm.

The correlation between the PWV values measured with the arteriograph and each of the fit parameters of the decay function, y(t), on the recovery stage, is summarized in Figure 4 and Table 2. Also, the correlation between PWV and the measured values of the heart rate at different time moments during the test (HR-Art, measured with the arterioraph; HR-Pre, measured at time t_2_ after the subject rose from the supine position to the orthostatic position, and HR-4min, after 4 min of moderate physical effort and HR-6min, after 6 min of relaxation in the recovery stage) are summarized in Figure 5 and Figure 6, as well as in Table 3. Also, Table 2 and Table 3 include the regression equations for pulse wave velocity as functions of the parameters y_o_, τ, and heart rate measured at various times during the test (HR-Art, HR-Pre, HR-4min, and HR-6min).

## 4. Discussion

European studies on PWV show that, for age groups under 30 years with optimal blood pressure values, the normal values of PWV range between 4.7 m/s and 7.5 m/s, with an average of 6.1 m/s [11]. From Table 1, it can be seen that, for our study group, the average value of PWV is 7.34 m/s and the standard deviation is 1.32 m/s, while their blood pressure values are in an optimal range. Taking into account the fact that our study group consisted of resident doctors and medical students who were practicing in a hospital, we consider that the high values of their PWV are due to daily mental stress as well as workplace stressors. We also consider that the stress felt by young people in their daily activity in the hospital during pandemic conditions (under which the study was conducted) should not be neglected. As a matter of fact, in a study carried out in the United States, the highest level of stress related to the COVID-19 pandemic was noted among young people [12]. The relationship between mental stress and PWV was investigated by various researchers, and it was reported that even a few minutes of mental stress led to increased PWV [13,14,15]. For instance, in the work of Kume et al. [15], even after only 5 min of mental stress, significant elevations of segmental PWV were recorded, and the changes persisted until 30 min in the group that simply rested on a chair after the mental stress test. A recent study [16] analyzed the association between PWV and general stress in a group consisting of medical staff. In this study, it was reported that working in a hospital during COVID-19 pandemic conditions led to increased PWV values compared to those measured in nonpandemic conditions.

During physical effort, several interrelated cardiovascular changes occur, with the primary goal of increasing blood flow to the working muscles [17]. Among these changes, heart rate is one of the simplest physiological responses. When the physical effort intensity is held constant, HR increases until it reaches a plateau, similar to the schematic representation from Figure 2. This plateau is the steady-state heart rate, meeting the circulatory demands at that specific constant effort. An increase in the intensity of physical effort at a new constant level leads to an increase in HR to a new steady-state value within 2 to 3 min [17]. When physical effort is reduced, HR decreases to meet the new circulatory demands. The plateau value for HR depends on the cardiorespiratory fitness of the subjects (those who are less fit have higher values of HR) [17]. On the other hand, it is well known that physically trained subjects have good arterial elasticity and low PWV values [18,19,20,21,22,23].

The fact that the evolution (over time) of HR after being suddenly interrupted by physical effort is of an exponential type (similar to any response function to a step stimulus—see Figure 2 and Figure 3), forcing us to apply the response function method in the assessment of arterial stiffness by correlating PWV with the parameters of the typical response function to a step stimulus, y(t) = y_o_ + A∙exp(−t/τ).

As mentioned in the materials and methods section, y_o_ is the resting heart rate value after physical effort, calculated by fitting it with the exponential decay function y(t). The analysis of the correlation between PWV and y_o_ showed a moderate and positive correlation (the larger y_o_, the larger PWV). Taking into account the maximum normal value of PWV in the European healthy population aged under 30 years and the average heart rate of the healthy population (in order of 80 b/min), the dependence of PWV on y_o_ can be divided into four quadrants (see Figure 4a). Also, we may observe from Figure 4a that 55.5% of the subjects have a normal PWV (smaller than 7.5 m/s) and an HR smaller than 80 b/min. In other words, among those subjects with normal PWV values, 90.9% have a y_o_ smaller than 80 b/min.

The analysis of the correlation between PWV and the relaxation time, τ, also revealed a moderate and positive correlation. The average value of the relaxation time, τ_ave_, is 0.64 min with a standard deviation of 0.39 min. If we set, as normal, the upper limit of the relaxation time of 1.03 min (τ_ave_
*+* standard deviation) and take the maximum normal value of PWV in the healthy European population aged under 30 years (i.e., 7.5 m/s) into account, the dependence of PWV on τ can be divided into four quadrants (see Figure 4b). We may observe from Figure 4b that 58.3% of the total subjects have normal PWV (smaller than 7.5 m/s) and a relaxation time smaller than 1 min. Also, among those subjects with normal PWV values, 95.4% have a τ smaller than 1 min.

The parameter, A, of the exponential decay function represents the difference between the heart rate at the beginning of the recovery stage and the resting heart rate after physical effort, and no correlation was found between PWV and A.

The concept of a steady-state heart rate forms the basis for simple exercise tests that have been developed to estimate cardiorespiratory fitness [17]. The subjects with a better cardiorespiratory endurance capacity will have a lower steady-state HR than those who are less fit [17]. Bearing in mind the correlation between good cardiorespiratory fitness and low PWV, apart from the parameters of the exponential decay function, y(t), a correlation analysis of PWV with the steady-state heart rate measured at various times during the test (HR-Art, HR-Pre, HR-4min, and HR-6min) was also performed.

The correlation analysis of PWV with steady-state HR values is summarized in Figure 5 and Figure 6, as well as in Table 3. We noticed a good correlation between PWV and three of the steady-state values of HR measured during the test and a weak correlation with HR-4min.

The best Pearson’s correlation coefficient was obtained for PWV dependence on heart rate, which was measured with the arteriograph in the supine position after at least 10 min of rest (HR-Art). A good correlation between the resting heart rate and PWV was also reported in references [24,25,26]. Taking into account the fact that, for the European population under the age of 30, the normal value of PWV is below 7.5 m/s [11], and the average normal value for HR is 80 b/min, Figure 5a can be divided into four quadrants. We have to remark (from Figure 5a) that among the subjects from the study, those who had a PWV within the normal limits (i.e., below 7.5 m/s) all had a heart rate below 80 b/min.

One of the parameters that can be easily evaluated for any subject (and which does not require physical effort testing equipment) is the value of HR measured after the subject got up from the supine position to the orthostatic position, HR-Pre. We must note that among those who have an HR-Pre larger than 100 b/min, all had PWV values greater than the normal upper limit for the age group under 30 years old (which is 7.5 m/s [11]).

The correlation of PWV with HR-6-min is similar to that between PWV and y_o_. This fact was expected because y_o_ and HR-6min represent the steady-state HR measured after 6 min of rest in the supine position. The difference between y_o_ and HR-6min consists in the fact that HR-6min is a measured value, whilst y_o_ is a value resulting from a fit. We remind the reader here that the fit can be interpreted as a form of smoothing of the experimental dependencies. In other words, y_o_ can be seen as an average, and therefore, the Pearson’s coefficient for the correlation of y_o_ with PWV is lower than the Pearson’s coefficient for the correlation between HR-6min and PWV.

The weakest correlation between PWV and the steady-state HR values during the test was recorded in the case of HR-4min, which is the HR value measured after 4 min of moderate physical effort (walking on the treadmill at the speed of 2.7 km/h).

During any prolonged physical effort, one of the cardiovascular responses is vasodilation [27]. For the same necessary volume of blood in the muscles and tissues, the required heart rate is lower in dilated vessels than in nondilated vessels. On the other hand, PWV is measured in the supine position after at least 10 min of relaxation, so the blood vessels are not dilated by physical effort when the PWV is measured. This could be a possible explanation for a lower correlation between PWV and HR-4min than between PWV and HR-Art. Another reason for the weak correlation between PWV and HR-4min is related to the fact that, during physical effort, blood pressure increases, and the speed of the pulse wave increases with the increase in blood pressure [28,29]. Therefore, PWV measured with the arteriograph in the supine position is different from the PWV corresponding to high blood pressure during the physical effort.

The results of this study allow us to highlight some of the limits, which, if exceeded, can be associated with high PWV values:After a moderate physical effort and a resting time of at least 6 min, the HR remains at values larger than 80 b/min;The relaxation time, τ, in a physical effort test of moderate intensity is larger than 1 min;The HR measured after the subject is raised from the supine position to the orthostatic position is larger than 100 b/min;Resting HR is larger than 80 b/min.

However, these must be seen as preliminary results. For their possible use in clinical practice, validation will be needed for a large number of subjects and on all age groups, for example, in a meta-study regarding the normal limits of PWV [11].

## 5. Conclusions

The study shows that the use of step function physical effort as a stimulus and the recording of time-dependent HR can be used for the preliminary assessment of PWV.

PWV correlates with the steady-state values of HR, both in response to a minimal effort (such as the change from the supine to orthostatic position) as well as with the steady-state values of HR during physical effort or after relaxation in the supine position. Relaxation time after moderate physical effort (such as walking on a treadmill) is another parameter that correlates with PWV.

The analyzed parameters allowed for the highlighting of some normal limits within the investigated parameters, beyond which PWV has higher values than those considered normal for the analyzed age group.

## Figures and Tables

**Figure 1 medicina-58-01633-f001:**
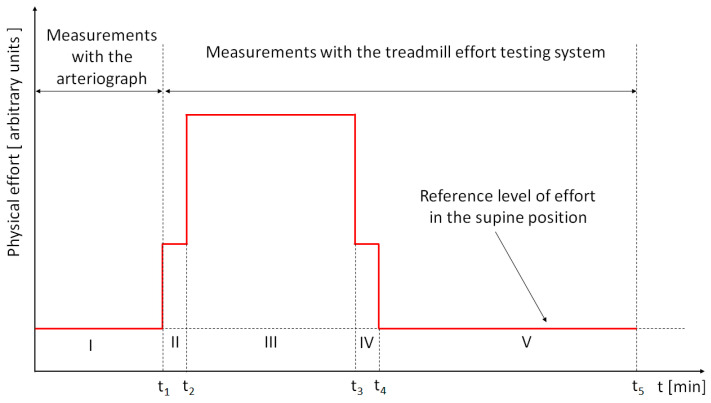
The stages of the measurements and the time dependence of the physical effort in arbitrary units: I—measurements with the arteriograph, II—raising the subject from the supine position and positioning on the treadmill, III—the loading stage, IV—the cooling stage and placing the subject in supine position for relaxation, and V—recovery stage. The red line represents physical effort, as a function of time, in arbitrary units.

**Figure 2 medicina-58-01633-f002:**
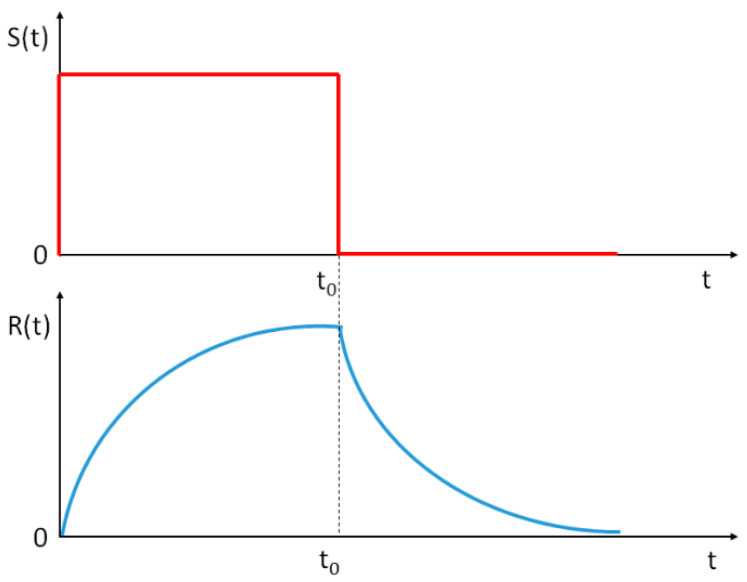
Theoretical representation of the time dependence of the response quantity, R(t), following a change as a step function of the stimulus quantity, S(t).

**Figure 3 medicina-58-01633-f003:**
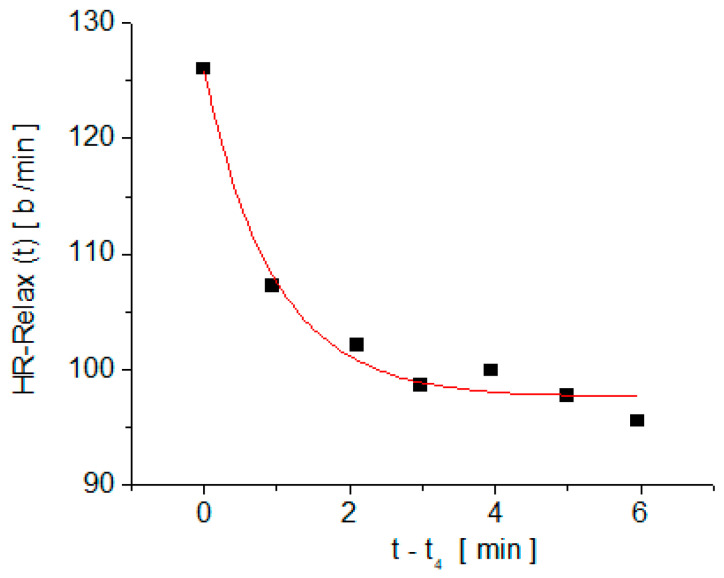
Typical time dependence of the heart rate measured in stage IV; squares are the measured values, and the full line is the fit with the exponential decay function y(t) = y_o_ + A∙exp(−t/τ).

**Figure 4 medicina-58-01633-f004:**
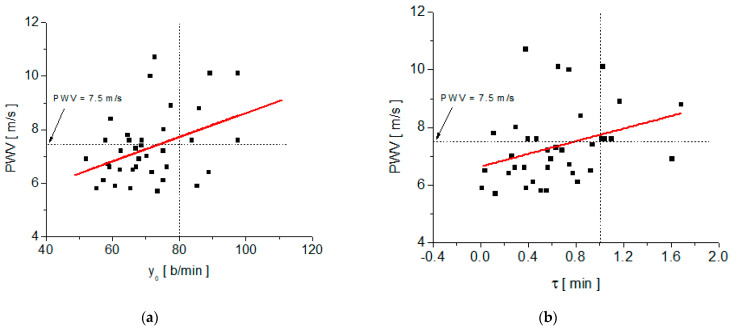
Plots of PWV versus the parameters y_o_ and τ and their linear fit: (**a**) PWV versus y_o_; (**b**) PWV versus τ.

**Figure 5 medicina-58-01633-f005:**
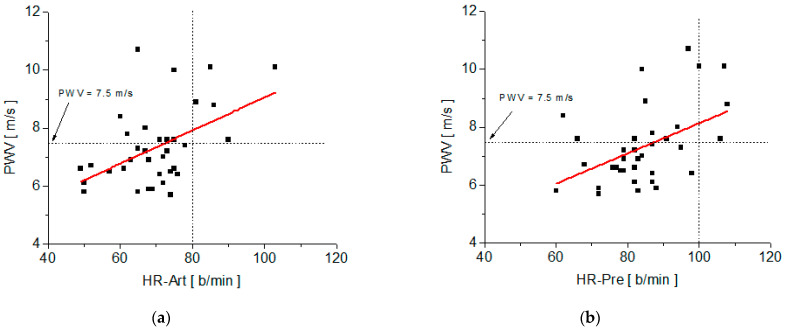
Plots of PWV versus heart rate measured at various times of the test and their linear fit: (**a**) PWV versus HR-Art; (**b**) PWV versus HR-Pre.

**Figure 6 medicina-58-01633-f006:**
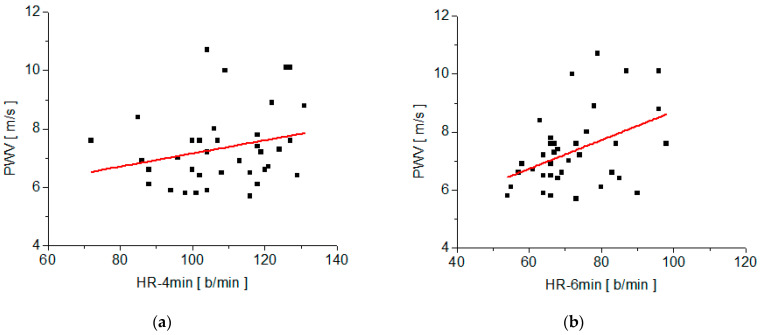
Plots of PWV versus heart rate measured at various times of the test and their linear fit: (**a**) PWV versus HR-4min; (**b**) PWV versus HR-6min.

**Table 1 medicina-58-01633-t001:** Characteristics of the subjects in the study group.

Parameter	Mean Value	Standard Deviation
Age	26.27 years	3.23 years
Anthropometric measurements		
Body Height	1.71 m	0.09 m
Body Weight	66.72 kg	15.89 kg
Body Mass Index	22.76 kg/m^2^	3.97 kg/m^2^
Arteriograph measurements		
Systolic BP (SBP-Art)	116.05 mmHg	9.20 mmHg
Diastolic BP (DBP-Art)	67.72 mmHg	8.12 mmHg
Heart rate (HR-Art)	69.91 b/min	11.31 b/min
PWV	7.34 m/s	1.32 m/s

**Table 2 medicina-58-01633-t002:** Summary of the correlation between PWV and the parameters y_o_, A, and τ.

	Regression Equation	Pearson’s Correlation Coefficient	Significance ^1^
Correlation of PWV with y_o_	PWV = 4.130 + 0.045∙y_o_	0.395	0.017
Correlation of PWV with τ	PWV = 6.640 + 1.097∙τ	0.329	0.049
Correlation of PWV with A	-	−0.090	0.599

^1^ for correlation, a two-tailed test of significance was used.

**Table 3 medicina-58-01633-t003:** Summary of the correlation between PWV and the heart rate measured at various times of the test.

	Regression Equation	Pearson’s Correlation Coefficient	Significance ^2^
Correlation of PWV with HR-Art	PWV = 3.353 + 0.057∙HR-Art	0.490	0.002
Correlation of PWV with HR-Pre	PWV = 2.932 + 0.052∙HR-Pre	0.479	0.003
Correlation of PWV with HR-4min	PWV = 4.907 + 0.022∙HR-4min	0.245	0.150
Correlation of PWV with HR-6min	PWV = 3.774 + 0.049∙HR-6min	0.433	0.008

^2^ for correlation, a two-tailed test of significance was used.

## Data Availability

The study protocol information will be provided upon request.

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
