# Peer review of "A Heart Rate Step Function Response Method for the Evaluation of Pulse Wave Velocity as a Predictor of Major Adverse Cardio-Vascular Events"

_medicina, 2022, doi:10.3390/medicina58111633_

Round 1
Reviewer 1 Report
the study was interesting and well done. small changes should be suggested. Remove the correlation and significant values from the discussion (results contain these); remove the Pearson's coefficient or R-square from the tables; there was not association with HR 4min: could you explain the reason?.
Reviewer 2 Report
The article can be improved in the following points:
1) It is not clear why you use the analogy with studies in physics in introduction part, if the concept of functional tests and functional diagnostics is commonly used in physiology and medicine for many years. Many studies with different diagnostic directions showed that analyses of the dynamics of physiological processes (from rest to test) is more effective than just rest state parameters. Also, such approach is often considered as a way to offset the influence of the initial individual differences. In introduction, it would be better to discuss these methods in physiology and to highlight the novelty of guessed in the article functional tests.
2) Th study was based on the data of healthy people and performed correlation between HR parameters and PWV. The presented correlations and R2 values are significant but not strong, what can let us suggest that the same level of correlations and regression would be got for sample of people with different cardiovascular diseases? Why don't you expect the lost of relations between HR and BW under cardiovascular conditions?
3) If you use parametric statistical procedures (Pearson correlation and Linear regression) you should add information about normality of distribution of variables that you use in analysis.
4) In method part it would rather to add more details about ECG and PW registration protocols: time resolution of signals, applied filters, etc.
5) There is a misprint in discussion part - COVIV-19
